# Exploring Pt-Pd Alloy Nanoparticle Cluster Formation through Conventional Sizing Techniques and Single-Particle Inductively Coupled Plasma—Sector Field Mass Spectrometry

**DOI:** 10.3390/nano13182610

**Published:** 2023-09-21

**Authors:** Omar Martinez-Mora, Kristof Tirez, Filip Beutels, Wilfried Brusten, Luis F. Leon-Fernandez, Jan Fransaer, Xochitl Dominguez-Benetton, Milica Velimirovic

**Affiliations:** 1Flemish Institute for Technological Research (VITO), Boeretang 200, 2400 Mol, Belgiumkristof.tirez@vito.be (K.T.); filip.beutels@vito.be (F.B.); wilfried.brusten@vito.be (W.B.); luis.leon@vito.be (L.F.L.-F.); xochitl.dominguez@vito.be (X.D.-B.); 2Department of Materials Engineering, Surface and Interface Engineered Materials, Katholieke Universiteit Leuven, Kasteelpark Arenberg 44—Box 2450, 3001 Leuven, Belgium; jan.fransaer@kuleuven.be

**Keywords:** Pt nanoparticles, Pd nanoparticles, Pt-Pd alloy nanoparticles, gas diffusion electrocrystallization, high-resolution single-particle inductively coupled plasma-mass spectrometry, scanning electron microscopy, dynamic light scattering

## Abstract

Accurate characterization of Pt-Pd alloy nanoparticle clusters (NCs) is crucial for understanding their synthesis using Gas-Diffusion Electrocrystallization (GDEx). In this study, we propose a comprehensive approach that integrates conventional sizing techniques—scanning electron microscopy (SEM) and dynamic light scattering (DLS)—with innovative single-particle inductively coupled plasma—sector field mass spectrometry (spICP-SFMS) to investigate Pt-Pd alloy NC formation. SEM and DLS provide insights into morphology and hydrodynamic sizes, while spICP-SFMS elucidates the particle size and distribution of Pt-Pd alloy NCs, offering rapid and orthogonal characterization. The spICP-SFMS approach presented enables detailed characterization of Pt-Pd alloy NCs, which was previously challenging due to the absence of multi-element capabilities in conventional spICP-MS systems. This innovative approach not only enhances our understanding of bimetallic nanoparticle synthesis, but also paves the way for tailoring these materials for specific applications, marking a significant advancement in the field of nanomaterial science.

## 1. Introduction

Over the past years, Pt and Pt-based nanoparticles (NPs) have emerged as highly valuable materials with a particular emphasis on their applications in fuel cells, especially as electrocatalysts for the hydrogen oxidation reaction (HOR) [1], the hydrogen evolution reaction (HER) [2], the oxygen reduction reaction (ORR) [3], or the methanol oxidation reaction (MOR) [4]. However, the high price and risk of supply interruptions of Pt have prompted the exploration of alternatives to substitute and minimize the Pt quantity while improving the stability and activity of the electrocatalyst materials [5]. Alloying Pt with other metals has emerged as a promising strategy, aiming to accomplish both goals while improving catalytic performance. Among several candidates, Pd stands out due to its identical crystal structure and lattice constant comparable to that of Pt. This results in single-phase bi-metallic materials with strong coupling and improved catalytic performance attributed to changes in the electronic structure, promoting a better interaction with reaction intermediaries [6].

While the idea of using Pd might seem counterintuitive given its higher cost and limited availability, it is essential to note that the goal here is not solely to substitute Pt but to optimize the balance between performance and material usage. The intrinsic properties of Pd, the potential for enhancing catalytic performance, and the synergy with Pt in alloy formation make it a compelling choice despite its elevated cost. Exploring Pt-Pd alloy nanoparticles becomes an exercise in achieving the best possible compromise between efficiency, cost-effectiveness, and the sustainable use of critical raw materials.

The catalytic performance of Pt-Pd NPs is influenced by their size, structure, and shape, which can be finely controlled through various synthesis parameters. For instance, the manipulation of precursor concentrations, the choice of stabilizers or reaction media, and the precise control of temperature, among other factors, have crucial roles in this tuning process, [7]. Taking into account these points, we have employed Gas-Diffusion Electrocrystallization (GDEx), an electrochemical technique, to synthesize noble metal NCs. This process utilizes in situ electrochemical reduction reactions, yielding gases which further act as reducing agents for noble metal ions, leading to the formation of small metal nanoclusters (20–30 nm) composed of much smaller primary nanoparticles (2–5 nm) [8,9,10].

The characterization of bimetallic Pt-Pd NPs requires diverse techniques, from conventional methods such as scanning/transmission electron microscopy (S/TEM) and dynamic light scattering (DLS) to advanced ones such as single-particle inductively coupled plasma mass spectrometry (spICP-MS) and single-particle time-of-flight ICP-MS (spTOF-ICP-MS) [11,12,13,14].

spICP-MS stands out due to its capability of discerning between ionic and particulate signals, providing a comprehensive analysis of NPs with minimal preparation and superior sensitivity [15,16]. However, the utilization of spICP-MS for complex samples, including bimetallic NPs, remains limited due to the restricted multi-element capabilities or their absence altogether when employing quadrupole-based ICP-MS systems, which are the most commonly used mass analyzers in ICP-MS instrumentation [12]. Nevertheless, by analyzing the signals from single particles, spICP-MS can provide information about the metal content and distribution within metallic NPs, even at very low concentrations, especially with the advent of single-particle inductively coupled plasma–sector field mass spectrometry (spICP-SFMS) instruments, which have significantly enhanced detection capabilities compared to other types [17]. Finally, these characterization techniques, when used in combination, can provide a comprehensive understanding of the size, morphology, aggregation state, and elemental composition of Pt-Pd bimetallic nanoparticles.

Our objective was to characterize Pt-Pd alloy NCs, using a blend of conventional techniques (SEM and DLS) and the advanced spICP-SFMS. We synthesized different Pt-Pd alloy NPs (Pt_100_, Pt_75_-Pd_25_, Pt_50_-Pd_50_, Pt_25_-Pd_75_, and Pd_100_) with varying Pt and Pd ratios through the GDEx process, adding polyvinyl pyrrolidone (PVP) as a stabilizer for preventing diffusion-limited aggregation [18]. This study aims to understand the synthesis and characterization of these bimetallic nanoparticles at the individual particle level, focusing on their elemental composition, size distribution, and formation mechanisms. Our results showcase that spICP-SFMS provides a comparable, if not superior, characterization of Pt-Pd alloy NCs as traditional techniques, offering added advantages like detecting low concentrations and distinguishing elemental states.

## 2. Materials and Methods

### 2.1. Chemicals and Materials

All chemicals used in this study were of analytical-grade purity. For spICP-SFMS analysis, ultra-pure water with a resistance of 18.2 MΩ cm was obtained using a Milli-Q system (Millipore, Billerica, MA, USA). High-purity (optima grade) 14 M HNO_3_ and 12 M HCl were procured from Fisher Chemical (Loughborough, UK). To develop the method and calibrate the spICP-SFMS instrument (Nu Attom, Nu Instruments Ltd., Wrexham, UK), appropriate dilutions of 1000 mg L^−1^ Au Certipur^®^ (Merck, Darmstadt, Germany) in 2 M HNO_3_ traceable to SRM from NIST H(AuCl_4_) were used. Suspensions of spherical gold nanoparticles (GNPs) with a diameter of 27.6 (NIST SRM 8012) and 56.0 nm (NIST SRM 8013) from the National Institute of Standards and Technology (NIST, USA) [19,20] were used to determine the transport efficiency (TE) based on the particle size method [21].

### 2.2. Synthesis of Bimetallic Pd-Pt Alloy NCs

The previously described GDEx electrochemical setup was employed to synthesize the Pt-Pd alloy NPs [8,9,10]. The setup consisted of an EC Micro Flow Cell (ElectroCell, Tarm, Denmark) with a three-compartment design. A VITO CORE^®^ gas-diffusion electrode (GDE) was employed as a cathode. A leak-free Ag/AgCl reference electrode (eDAQ, Colorado Springs, CO, USA) was placed near the GDE. As for the anode, a tantalum plate coated with platinum (10 μm thickness) was utilized. The exposed geometric area of the cathode and anode was 10 cm^2^, respectively. The anodic and cathodic chambers had a thickness of 0.4 cm and a total chamber volume of 4 mL. Both chambers were separated by an ion-permeable porous separator (ZIRFON^®^ PERL, Agfa Gevaert, Mortsel, Belgium). CO_2_ was supplied at the hydrophobic (back) side of the GDE at a flow rate of 200 mL min^−1^ with an overpressure of 20 mbar(g).

Different catholyte solutions were prepared by varying the concentration of H_2_PtCl_6_ and PdCl_2_, as reported in Table 1, with 0.5 M of NaCl and 1 g L^−1^ of PVP. The pH was adjusted to 3 using HCl or NaHCO_3_ if necessary. A 0.5 M NaCl solution was used as the anolyte. Then, 100 mL of the catholyte and anolyte were added to two 3-necked glass bottles and connected to the electrochemical reactor using marprene tubing (Watson-Marlow, Falmouth, UK). The catholyte and anolyte solutions were pumped to their respective chambers at a flow rate of 100 mL min^−1^ using a peristaltic pump (530, Watson-Marlow, UK). The solutions were recirculated for 30 min before starting the experiments (without electrode polarization, OCV conditions) to ensure sufficient electrode wetting. Chronopotentiometric (CP) experiments were carried out in batch mode at −30 mA cm^−2^ using a VMP3 Bio-Logic (Claix, France) multichannel potentiostat. pH, charge, and potential were monitored throughout all experiments. The pH was measured every 5 s with a pH/ion meter (781, Metrohm, Herisau, Switzerland) equipped with a Metrohm Unitrode pH electrode. To monitor the metal concentration in the liquid phase, aliquots of 1 mL were taken from the catholyte at different times, centrifuged, and the supernatant analyzed with an inductively coupled plasma-optical emission spectrometer (ICP-OES) (750 ES, Varian, Palo Alto, CA, USA).

After the synthesis, to remove the NaCl, the different Pt-Pd alloy NCs suspensions were dialyzed using a dialysis membrane (VWR, 12–14 kD). This process was repeated until no chlorides could be detected in the solution using AgNO_3_ solution (if chlorides are present in the medium, they will precipitate the Ag+ ions as AgCl). The Pt and Pd concentration in the different NPs suspensions was measured with ICP-OES after digestion with aqua regia. All Pt NPs dispersions were kept in closed glass vessels and stored in the dark until further processing.

### 2.3. Characterization of Bimetallic Pd-Pt Alloy NCs

#### 2.3.1. SEM

SEM images were obtained using a Philips XL30 FEG (Eindhoven, The Netherlands) scanning electron microscope with secondary electrons and an acceleration voltage of 30 kV. The samples were prepared by placing two drops of the dialyzed suspensions on an aluminum foil mounted on a sample holder. The mean particle size and distribution were evaluated by counting more than 100 particles using the image processing software (1.53t), ImageJ. After that, data were fitted to a lognormal distribution to obtain the mean particle size and standard deviation.

#### 2.3.2. DLS

A Zetasizer Nano ZS (Malvern Panalytical Ltd., Malvern, UK) was used to perform Dynamic Light Scattering (DLS) measurements on colloidal suspensions. The concentrated samples were diluted in DI water to the range of 0.1 g mL^−1^ particles to water. A refractive index of 2.320 (for Pt) and 1.770 (for Pd), and an extinction coefficient of 4.160 (for Pt) and 4.290 (for Pd) were used for DLS measurements. All determinations were repeated in triplicate, with at least 3 measurements recorded for each run.

#### 2.3.3. spICP-SFMS

The Nu Attom ICP-MS instrument (Nu Instruments Ltd., Wrexham, UK) was employed for the analysis of all samples. This instrument features a double-focusing sector field mass spectrometer utilizing forward (Nier-Johnson) geometry. In the single-particle mode, a specific *m*/*z* value was monitored with consistent magnetic field and acceleration voltage settings. For the characterization of Pd/Pt nanoparticles (NPs), the instrument was operated at low resolution (R ∼ 300). Sample introduction was accomplished using a conventional system comprising a glass concentric nebulizer with a nominal uptake rate of 300 μL min^−1^ coupled to a quartz cyclonic spray chamber. The precise uptake rate was determined by measuring the water mass before and after transferring the sample into the system via the peristaltic pump over a duration of 10 min. The spICP-SFMS analysis was performed in time-resolved analysis (TRA) mode, with a dwell time of 50 µs and an acquisition time of 60 s (Table 2). Nebulization efficiency was determined using the spherical gold nanoparticle standards as described in Section 2.1.

Data acquisition and data treatment were performed using the combination of NuAttoLab v2.11.0.137 and NuQuant v2.2.2383.3 software (Nu Instruments, Wrexham, UK). The characterization of bimetallic nanoparticles in this study was conducted using spICP-SFMS. According to the fundamental principle of spICP-SFMS, the diluted sample solution containing Pt-Pd alloy NCs is introduced into the plasma, resulting in the generation of ion clouds composed of the two different metals. The mass spectrometer, functioning as a mass analyzer, detects the *m*/*z* (mass-to-charge ratio) of the analyte sequentially. The intensity of each spike pulse, assumed to originate from a single particle, is subsequently correlated with the particle’s mass (*m*_p_). Subsequently, the particle diameter (*d*) can be determined using the following calculation method [12]:$$d=\sqrt[{3}]{\frac{6m_{p}}{\rho \pi }}$$

The data acquired through spICP-SFMS provide an equivalent spherical size. In the case of Pt-Pd alloy NPs, the particle size can be estimated based on the equivalent spherical size of Pd and Pt (Figure 1) utilizing the particle density ($\rho_{\mathrm{Pt}}=21,450\;\mathrm{kg}\cdot m^{-3};\;\rho_{\mathrm{Pd}}=12,023\;\mathrm{kg}\cdot m^{-3}$). Initially, the equivalent sizes of the two metal NPs, assuming a spherical shape, are converted into volumes (V_Pd_ and V_Pt_). Subsequently, the total volume was calculated by summing up the converted volumes, and the resulting value was used to determine the equivalent spherical diameter of the Pd-Pt bimetallic alloy NCs.

## 3. Results and Discussion

### 3.1. Synthesis of the Pt NCs, Pd NCs, and Pt-Pd Alloy NCs

In Figure 2, the pH and the evolution of metal concentration as a function of time and volumetric charge are presented, and a blank experiment (without metals) is added as a comparison. With the GDEx process, the reagents for forming metallic NPs are produced in situ by reducing a gas (i.e., CO_2_) on a gas diffusion electrode. At the synthesis conditions, the product of the CO_2_ reduction at the GDE in aqueous electrolyte is mainly CO (reaction 1); moreover, the H_2_O reduction to H_2_ occurs simultaneously (reaction 2) [22].
CO_2_ + 2H_2_O + 2e^−^ ⇌ CO + 2OH^−^, E^0^ = −0.934 V_SHE_(1)
2H_2_O + 2e^−^ ⇌ H_2_ + 2OH^−^, E^0^ = −0.828 V_SHE_(2)

Both reactions generate OH^−^, leading to the pH increase of the catholyte, as can be observed in the blank experiment without Pt and Pd ions. At the same time, the dissolution of unreacted CO_2_ and its chemical equilibrium to ${HCO}_{3}^{-}$ and ${CO}_{3}^{2-}$ (reactions 3 and 4) take place, consuming OH^−^ and buffering the pH rise. Consequently, CO_2_ equilibrium is crucial to avoid the precipitations of the dissolved metal ions (especially Pd) as hydroxides.
CO_2_ + OH^−^ ⇌ HCO_3_^−^(3)
HCO_3_^−^ + OH^−^ ⇌ CO_3_^2−^ + H_2_O(4)

A fraction of the produced H_2_ and CO is transported to the bulk electrolyte [9], where H_2_ drives the reduction of the Pt(IV) and Pd(II) ions in the solution and the formation of the Pt-Pd alloy NPs, while CO acts as a size controller agent [8]. The reduction of Pt(IV) occurs in a two-step process (reactions 5 and 6) in which Pt(IV) (in the form of [PtCl_6_]^2−^) is first reduced to Pt(II) ([PtCl_4_]^2−^) and then further reduced to Pt^0^.
[PtCl_6_]^2−^ + H_2_ ⟶ [PtCl_4_]^2+^ + 2H^+^ + 2Cl^−^(5)
[PtCl_4_]^2−^ + H_2_ ⟶ Pt^0^ + 2H^+^ + 4Cl^−^(6)

The reduction of Pt(II) to Pt^0^ only occurs when approximately 90% of Pt(IV) has been reduced. This can be seen in Figure 2a, where the depletion of Pt ions in solution (formation of Pt NPs) starts after a certain period of polarization. However, as the reduction of Pt ions releases H^+^, a buffering of the electrolyte pH is observed. On the other hand, the reduction of Pd(II) ions (as [PdCl_4_]^2−^) occurs in one step (reaction 7). As seen in Figure 2a, the depletion of Pd ions and, hence, the formation of Pd NPs starts immediately after polarization.
[PdCl_4_]^2−^ + H_2_ ⟶ Pd^0^ + 2H^+^ + 4Cl^−^(7)

As observed with the individual metals, the reduction of Pd(II) is faster than Pt(IV); however, their depletion occurs simultaneously when both metals are present in the solution. This is more evident in Figure 2d, where the initial concentration of both metals is the same. This behaviour is explained by considering that upon the formation of Pd^0^, the galvanic displacement with Pt(II) occurs (reaction 8) [23]. This reaction catalyzes the formation of Pt^0^ but delays the formation of Pd^0^, resulting in the co-depletion of both metals and the formation of Pt-Pd alloys NCs.
Pd^0^ + [PtCl_4_]^2−^ ⟶ Pt^0^ + [PdCl_4_]^2−^(8)

### 3.2. Characterization of Pt NCs, Pd NCs, and Pt-Pd Alloy NCs Using SEM

Electron microscopy is considered one of the most powerful techniques for analyzing nanomaterials. It permits to obtain information on their size, shape, and aggregation state while also aiding in interpreting results derived from complementary analytical methods [24]. While transmission electron microscopy (TEM) is undeniably the most pertinent nanoparticle characterization technique, its cost and the need for extensively trained personnel cannot be overlooked. In that sense, scanning electron microscopy (SEM) has the advantage of being versatile, precise, easy to use, and widely available, and it is sufficient when analyzing larger nanoparticles [25]. In this regard, SEM was employed to comprehensively assess the particle size distribution of synthesized Pt NCs, Pd NCs, and Pt-Pd alloy NCs, contributing to an initial understanding of their formation processes. The obtained SEM images are presented in Figure 3, revealing that all the materials exhibited a nearly spherical particle shape. The calculated mean particle sizes were approximately 20 ± 4 nm for Pt NCs (Pt_100_) and 26 ± 16 nm for Pd NCs (Pd_100_). As observed, the mean particle size of Pd NCs is greater compared to Pt NCs. Furthermore, Pd NCs also exhibit a broader particle size distribution. This observation suggests that both metals have different nucleation and growth kinetics during synthesis. This can be attributed to the presence of CO. CO can be chemisorbed on the surface of both Pt and Pd clusters. In the case of Pt, this interaction blocks the nanoparticle surface, impeding particle growth and facilitating the formation of small nanoparticles characterized by a narrow size distribution. Conversely, CO can also reduce Pd(II) ions, which, unlike the Pt mechanism, promotes the Pd reduction kinetics, resulting in bigger nanoparticles with a broader size distribution. Extending the analysis to Pt-Pd alloy NCs, Pt_75_-Pd_25_, Pt_50_-Pd_50_, and Pt_25_-Pd_75_, the mean particle sizes were 20 ± 6 nm, 22 ± 9 nm, and 24 ± 12 nm, respectively. The clear trend of increasing particle size with higher Pd content suggests a potential correlation between composition and nanoparticle size.

Furthermore, the elemental mapping of the bimetallic NCs was performed using scanning electron microscopy coupled with energy-dispersive X-ray spectroscopy (SEM-EDS) and is presented in the Appendix A. The results show a homogeneous distribution of Pt and Pd in all three bimetallic compositions, confirming their alloy nature.

### 3.3. Characterization of Pt NCs, Pd NCs, and Pt-Pd Alloy NCs Using DLS

Even though SEM analysis can provide a meaningful analysis of the Pt-Pd alloy NCs, microscope images should always be supplemented with techniques that investigate a statistically more significant portion of the sample [7]. In that sense, DLS was employed as an effective and non-invasive screening technique to measure the hydrodynamic size of Pt, Pd, and Pt-Pd alloy NCs, thereby providing essential information on their size distribution and stability (Figure 4). This method holds particular significance in the initial screening phases of the synthesis process, where the rapid evaluation of particle size and stability is of paramount importance. The obtained DLS data for Pt NCs showed an average hydrodynamic size of 49 ± 18 nm with a Polydispersity Index (PDI) of 0.136, and 138 ± 54 nm with a PDI of 0.182 for Pd NCs. The DLS analysis further revealed hydrodynamic sizes of 85 ± 35 nm (PDI: 0.168), 94 ± 44 nm (PDI: 0.205), and 114 ± 47 nm (PDI: 0.182) for Pt_75_-Pd_25_, Pt_50_-Pd_50_, and Pt_25_-Pd_75_ alloy NCs, respectively.

The data show a much greater size distribution with DLS; however, it should be noted that DLS measurements include solvated and aggregate particles in the analysis that also contribute to the apparent size distribution. However, the trend observed with both techniques was similar; the particle size increases with the increase in Pd content. While SEM and DLS provide complementary insights into nanoparticle size characteristics, it becomes evident that careful consideration of the selected technique is essential when interpreting and reporting nanoparticle dimensions. The disparities observed between the SEM and DLS data underscore the importance of employing multiple analytical methods to comprehensively understand nanoparticle properties.

### 3.4. Feasibility of Using spICP-SFMS as a Complementary Technique to DLS and SEM

Although considerable progress has been made, spICP-MS is still an emerging analytical tool with numerous development opportunities [26]. spICP-SFMS stands as a robust avenue for NPs characterization and quantification, marked by its efficient sample preparation, remarkable sensitivity, and element-specific determination. However, these merits are accompanied by inherent constraints, prominently including potential limitations in multi-element analysis. The spherical equivalent diameter and particle size distribution data of Pt NCs, Pd NCs, and Pt-Pd alloy NCs were determined to examine the possibility of using spICP-SFMS for obtaining the particle size distribution of Pt-Pd alloy NPs. The Pt and Pd concentrations of Pt-Pd alloy NCs were determined in two runs: one for platinum and one for palladium (sequential analysis). spICP-SFMS measurements yielded an equivalent spherical diameter of 17.2 nm for Pt NPs (Pt_100_) and 37.4 nm for Pd NPs (Pd_100_) (Figure 5, Table 3). Figure 5 also shows a more homogenous particle size distribution for Pt_100_ compared to Pd_100_, which agrees with the observations from SEM analysis. Additionally, Pt_75_-Pd_25_, Pt_50_-Pd_50_, and Pt_25_-Pd_75_ alloy NPs exhibited a size of 30.7 nm, 29.0 nm, and 35.4 nm, respectively (Figure 6, Table 3). Furthermore, spICP-SFMS provides particle number concentration (Table 3). Finally, the particle sizes provided by SEM and spICP-SFMS are in good agreement, indicating that spICP-SFMS can be used as an alternative method to provide additional confirmation of Pt-Pd alloy NCs particle size distribution.

The differences observed between the three techniques can be attributed to the inherent nature of each method. SEM provides direct imaging of individual nanoparticles on a solid substrate, potentially leading to particle agglomeration or size overestimation due to sample preparation and imaging artifacts. On the other hand, DLS is a valuable tool for the fast determination of the hydrodynamic size distribution of nanoparticles in solution, despite its limitations related to sample types, particle shape, aggregation, and accuracy in size measurements. spICP-SFMS measures individual nanoparticles in solution, minimizing the effects of aggregation and offering a more accurate representation of the true particle size distribution. Finally, the synergistic application of used techniques has the potential to provide valuable insights into size-related aspects of the Pt NCs, Pd NCs, and Pt-Pd alloy NCs studied. (see Figure 5 and Figure 6).

## 4. Conclusions

A comprehensive characterization of GDEx-made Pt NCs, Pd NCs, and Pt-Pd alloy NCs employing an integrative approach that encompasses SEM, DLS, and spICP-SFMS has yielded valuable insights into their size distribution and morphological characteristics. The distinct variations in size observed among the Pt NCs and Pd NCs suggest the intricate interplay of nucleation, growth kinetics, agglomeration, and reduction kinetics during their synthesis processes. The spICP-SFMS analysis shows that it is possible to use spICP-MS for complex samples, including bimetallic Pt-Pd alloy NCs and offers a unique perspective, providing individual particle information and confirming the size trends observed through DLS and SEM for Pt NCs and Pd NCs. The observed increase in size with higher Pd content in Pt-Pd alloy nanoparticles measured via SEM and DLS indicates a potential composition–size correlation. This observation was less pronounced for spICP-SFMS. These findings enhance our understanding of bimetallic nanoparticle formation dynamics and pave the way for tailoring Pt-Pd alloy NCs for specific applications. Finally, this multidisciplinary approach provides a foundation for optimizing synthesis processes, achieving better control over nanoparticle properties, and advancing the design of Pt-Pd alloy NCs with tailored characteristics. 

## Figures and Tables

**Figure 1 nanomaterials-13-02610-f001:**
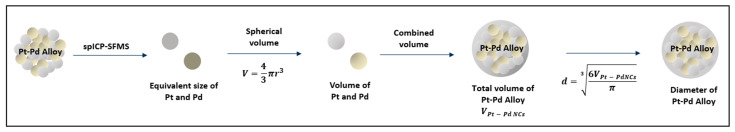
Schematic representation of the approach for calculation of particle size of Pt-Pd alloy NCs using the data obtained from spICP-SFMS.

**Figure 2 nanomaterials-13-02610-f002:**
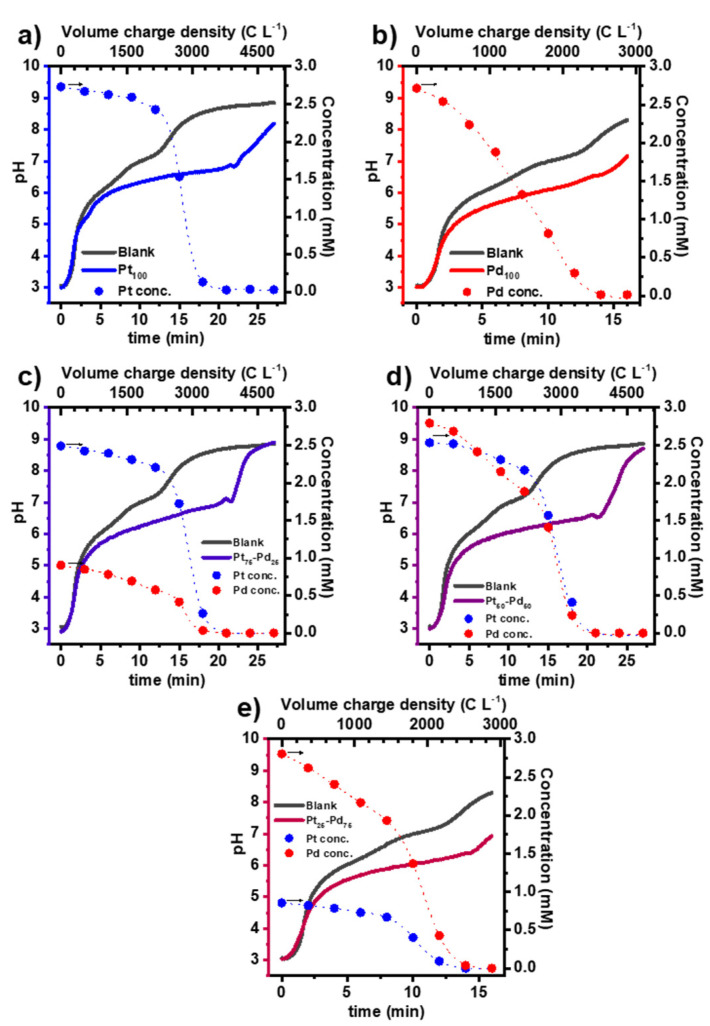
Evolution of pH and concentration of dissolved metals as a function of time and the charge consumed per unit volume of catholyte throughout the GDEx, leading to the synthesis of Pt NCs (**a**), Pd NCs (**b**), and Pt-Pd alloy NCs (**c**), Pt_75_-Pd_25_; (**d**), Pt_50_-Pd_50_; and (**e**), Pt_25_-Pd_75_).

**Figure 3 nanomaterials-13-02610-f003:**
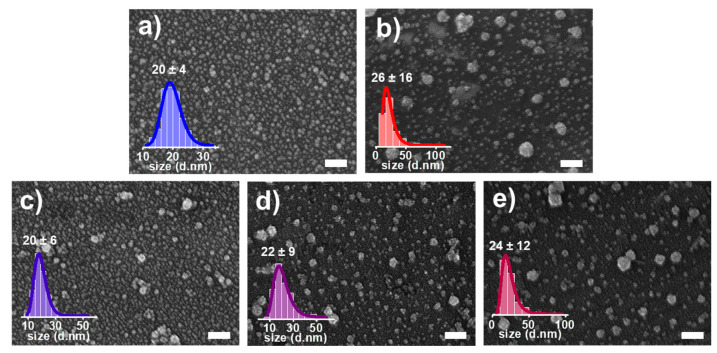
Particle size distribution of (**a**) Pt NCs, (**b**) Pd NCs, and Pt-Pd alloy NPs ((**c**), Pt_75_-Pd_25_; (**d**), Pt_50_-Pd_50_; and (**e**), Pt_25_-Pd_25_) as obtained using SEM. The white scale bar is 100 nm.

**Figure 4 nanomaterials-13-02610-f004:**
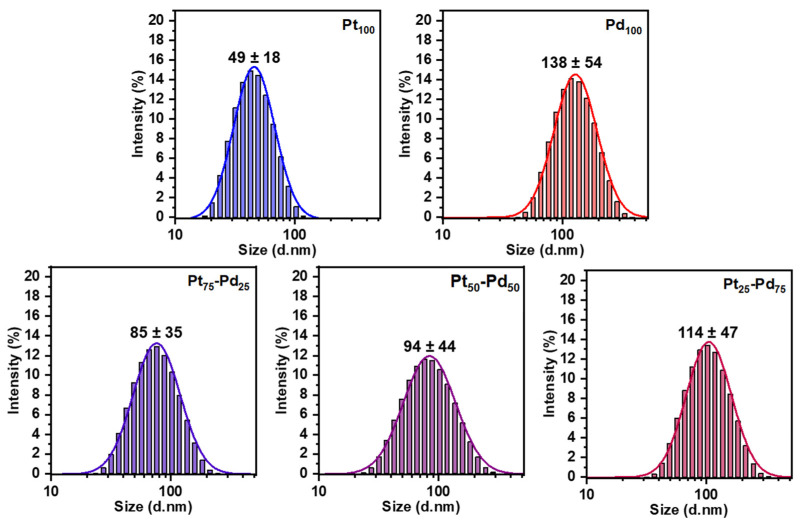
DLS measurements of the Pt NCs, Pd NCs, and Pt-Pd alloy NCs.

**Figure 5 nanomaterials-13-02610-f005:**
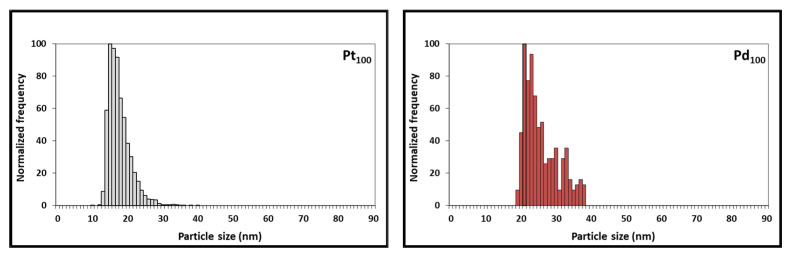
Particle size distribution of Pt (Pt_100_) and Pd (Pd_100_) NCs as obtained using spICP-SFMS.

**Figure 6 nanomaterials-13-02610-f006:**
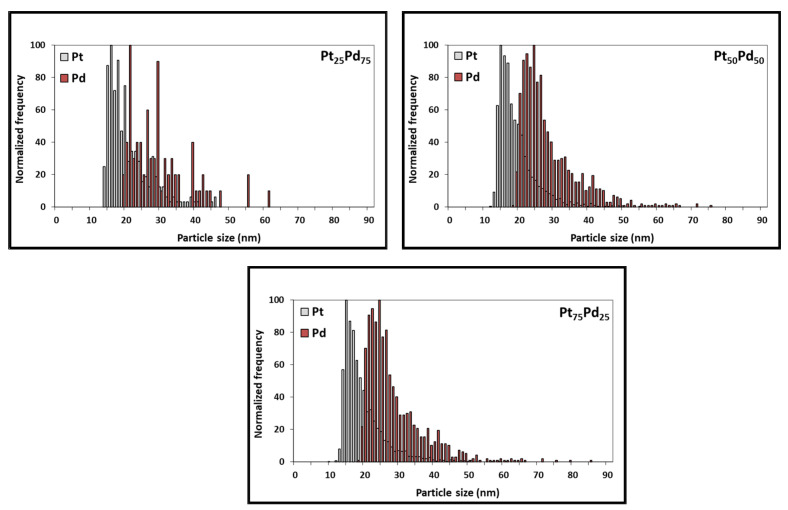
Particle size distribution of Pt and Pd present in Pt_75_-Pd_25_, Pt_50_-Pd_50_, Pt_25_-Pd_75_ alloy NCs as acquired through spICP-SFMS, used to determine the equivalent spherical diameter of the Pd-Pt bimetallic alloy NCs (see Table 3 and Appendix A).

**Table 1 nanomaterials-13-02610-t001:** Metal concentration of the catholyte solutions used to synthesize the Pt-Pd alloy NCs and the empirical and obtained composition of the synthesized materials.

Solution	Feeding		Composition	
	Pt(IV)	Pd(II)	Empirical	As Synthesized *
1	3 mM	--	Pt_100_	Pt_100_
2	3 mM	1 mM	Pt_75_-Pd_25_	Pt_74_-Pd_26_
3	3 mM	3 mM	Pt_50_-Pd_50_	Pt_54_-Pd_46_
4	1 mM	3 mM	Pt_25_-Pd_75_	Pt_26_-Pt_74_
5	--	3 mM	Pd_100_	Pd_100_

* Determined using ICP-OES.

**Table 2 nanomaterials-13-02610-t002:** ICP-MS instrument settings and data acquisition parameters.

Parameter	
Radio frequency power	1300 W
Plasma gas (Ar) flow rate	13 L min^−1^
Carrier gas flow rate	0.93 L min^−1^
Measurement mode	TRA
Nuclide monitored	^195^Pt, ^105^Pd
Type of detection for multiple elements	sequential
Dwell time	50 µs
Acquisition time	60 s
Nebulizer	MicroMist
Spray chamber	Cyclonic

**Table 3 nanomaterials-13-02610-t003:** Particle size of Pt NCs, Pd NCs and Pt-Pd alloy NCs using the data obtained from spICP-SFMS (N = 1).

	Spherical Equivalent Diameter (nm) *Normal Distribution **	Spherical Equivalent Diameter (nm) *Log Normal Distribution **	Spherical Equivalent Diameter (nm) *Cubic Spline **	Particle Number Concentration (Particles per mL) **
**Pt NCs** **(Pt_100_)**	17.2	16.4	16.0	3.70 × 10^5^
**Pd NCs** **(Pd_100_)**	37.4	26.0	22.2	2.84 × 10^4^
**Pt_75_-Pd_25_** **alloy NCs**	30.7	27.5	24.9	2.68 × 10^5^ (Pt-based) 8.99 × 10^4^ (Pd-based)
**Pt_50_-Pd_50_** **alloy NCs**	29.0	26.6	24.4	1.73 × 10^5^ (Pt-based) 2.54 × 10^4^ (Pd-based)
**Pt_25_-Pd_75_** **alloy NCs**	35.4	29.5	27.0	2.13 × 10^4^ (Pt-based) 2.44 × 10^4^ (Pd-based)

* Data fitting model; ** 1 × 10^6^ diluted samples with MilliQ water.

## Data Availability

The data presented in this study are available on request from the corresponding author.

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
