# Peer review of "Exploring Pt-Pd Alloy Nanoparticle Cluster Formation through Conventional Sizing Techniques and Single-Particle Inductively Coupled Plasma—Sector Field Mass Spectrometry"

_nanomaterials, 2023, doi:10.3390/nano13182610_

Round 1

Reviewer 1 Report

The manuscript presents interesting new results on the comparison of three methods for determining the size of nanoparticles of palladium, platinum, and their alloys of different composition. Of the three methods, two (SEM and DLS) are generally accepted and widely used, while the third is used much less frequently for these purposes, so its use gives the manuscript a significant novelty.

However, the manuscript evidently needs improvement. The abstract states that method spICP-SFMS allows a good understanding of the composition and the contribution of each of the metals to the volume and diameter of the bimetallic particle, and this is well depicted in Fig. 1. However, the results that would make it possible to visually confirm the proposed in Fig. 1 diagram, are not shown in the manuscript. I mean, the diagrams of particle size distributions for bimetallic particles, or any other data demonstrating the process, presented in Fig. 1, need to be added.

The other comments are presented below. They are presented in the order in which they appeared to the reviewer during the reading of the manuscript.

Comments.

11. The use of English and the designation of dimensions need careful checking. For example, for the first time I meet the word prowess in designating catalytic efficiency. On line 112 the exponent is not raised above the line in the designation of dimensions (1 g L-1 of PVP). Minutes or min, seconds or s?

2 2. In the heading of Table 1, it would be good to indicate by which method the real concentrations of metals in nanoparticles were determined.

33. Manufacturers and country of manufacture are not listed for all instruments used in the study (e.g. SEM, DLS)

44. The chapter numbering needs to be checked, and I would include the paragraph, which is numbered 2.3 in the version submitted to the reviewer, into the previous section, which is numbered 2.3.3.

55. Fig. 2 is of poor quality and needs improvement. Legends are unreadable. Besides, what is the concentration on Y axis? mM/L? In the figure caption it is said “and e, Pt25- Pd25). It needs correction.

66. In Fig. 3 the scale bar is presented on each image, but no data about the length of this scale.

77. When discussing the discrepancy in the results of the analysis of the sizes of nanoparticles by SEM and DLS, it would be better to say some words about the reasons for this discrepancy, instead of once again emphasizing the need of using several methods. Moreover, the discussion on this subject provided in the chapter 3.4 is also unsatisfactory and too general, not concerning the real properties of the systems under study. E.g. it is said about overestimation of particle sizes by SEM, whereas overestimation in reality is characteristic for DLS method. Why the results of SEM and spICP-SFMS are nearly the same for Pt particles and very different for Pd ones?

88. On fig. 4 it is desirable to indicate the particle size characterizing the positions of the peak maxima.

99.  I didn’t find any mention about the total number of particles which were measured by spICP-SFMS and included in diagrams in Fig. 5, as it was done when presenting the SEM results.

110.  I didn’t find the mention about total number of particles which were measured by spICP-SFMS and included in diagrams in Fig. 5, as it was done when presenting SEM results.

111.  It will be interesting to see the results of diagrams of particle size distribution for bimetallic particles provided by spICP-SFMS. I mean, may be the results for Pd and Pt can be joined on one graph for each bimetallic system?

see above

Author Response

Document is enclosed. 

Reviewer 2 Report

The article "Exploring Pt-Pd Alloy Nanoparticle Clusters Formation through Conventional Sizing Techniques and Single Particle Inductively Coupled Plasma - Sector Field Mass Spectrometry" presents a study in the field of obtaining and studying the structure of Pt-Pd nanoparticles of various compositions.

Several remarks should be noted:

Why the XRD method and the analysis of crystallite sizes according to Scherrer equation, the analysis of the lattice parameters of alloys of various compositions were not used in the work. This method can provide a lot of additional information about the structure of the resulting nanomaterials.

The formation of bimetallic particles has not been proven in the work, it is necessary to use XRD, possibly HRTEM, EDX line scanning, elemental mapping.

In figure 3, the size ruler is not visible.

Author Response

Document enclosed

Reviewer 3 Report

This is a very interesting work on the use of diverse characterization techniques for the proper evaluation of Pt-Pd alloy nanoparticle sizes. Especially spICP-SFMS comes up as an ‘advanced’ type of spICP-MS, with increased detection capability. The authors used an environmentally benign and very promising electrified technique, GDEx, for NP synthesis. The manuscript is very well written and I strongly recommend its acceptance, it will be an important addition to the literature. Some remarks to be addressed of rather ‘minor’ nature follow:

Introduction, line 36:

Regarding Pt and Pt-based NPs for fuel cells (EOR) as well as HER and ORR reactions, you can also check the following papers: Nanoscale, Vol 5, page 4776, year 2013 and Advanced Sustainable Systems, Vol 6, 2200163 (2022).

Line 70: Perhaps you can replace ‘However’ by ‘Nevertheless’ since also the previous phrase at line 67 starts with the same word (‘however’).

Line 81: For the multiple roles of PVP in nanoparticle synthesis (one of which is ‘stabilizer’), you can also check: Dalton Trans. 44, 17883 (2015).

Line 111: Maybe you can change ‘different concentrations’ to ‘varying concentrations’ in order to avoid the repetition of the word ‘different’ twice in the same phrase.

Lines 150-151: The refractive indices and extinction coefficients were set separately in the device only during the measurements of the monometallic samples? I assume that they were used also for the bimetallic particles. Therefore in the case of the bimetallic alloy samples, was it required to know in advance the Pt/Pd ratio in each sample before configuring the DLS device with the above indices of both metals? Is it required to set the Pt/Pd ratio also in the software?

Line 179: In that case, ‘a single particle’ refers to the metal nanoclusters of 20-30 nm size or to the primary NPs that constitute those bigger clusters?

Line 188: ‘the total volume was calculated by summing up the converted volumes’. This means that you somehow find an ‘average’ (or 'normalized') volume in the alloy considering the different volumes of the two metals? In that case, similarly to the measurement of DLS, a question comes up: Is it necessary to know in advance the Pd/Pt ratio in each sample to find such ‘converted’ (or ‘combined’) volumes, for a proper determination of the particle sizes?

Lines 217-218: For the role of H2 and CO gases in precursor reduction and in the control of size (and shape) of diverse types of nanostructures, including noble metal ones, you can check the following book chapter:

https://books.rsc.org/books/edited-volume/912/chapter-abstract/708857/Gases?redirectedFrom=fulltext

CO can interact strongly with metal surfaces, with chemisorption behavior which differs not only from one metal to another, but also from one crystallographic surface to the other for the same metal.

Line 238: The alloy formation is also facilitated by the very low lattice mismatch between the two metals (less than 1%). As you mention at line 41, they have a comparable lattice constant.

Line 264: The presence of CO plays its role for sure but the different behavior of the two metals in what concerns their nucleation and growth kinetics might be also partly due to their distinct reduction potentials? You mention something relevant at line 232.

Lines 323-333: Indeed, each characterization technique has its advantages and perhaps some shortcomings. That is why a complementary characterization with several techniques is very often beneficial to get the ‘whole picture’ for a given sample in the most reliable and precise way. You can check a very relevant Review paper: Nanoscale, 10, 12871 (2018). This review covers also SEM, DLS and spICP-MS, and it has a separate sub-section focusing on the measurement of NP size with different techniques.

The authors are not strongly recommended to necessarily cite all the above manuscripts and chapters. First of all they can check them to see if they find them useful for their work, citations can be only by choice.

Lines 415 and 417: Please correct ‘Nanopar-ticles’ to ‘Nanoparticles’.

Author Response

Document enclosed

Round 2

Reviewer 2 Report

The authors responded to the comments and the article can be accepted for publication.